# Changes of Somatosensory Phenotype in the Course of Disease in Osteoarthritis Patients

**DOI:** 10.3390/ijerph17093085

**Published:** 2020-04-29

**Authors:** Johanna Höper, Lara Schraml, Janne Gierthmühlen, Stephanie M. Helfert, Stefanie Rehm, Susanne Härtig, Ove Schröder, Michael Lankes, Frieder C. Traulsen, Andreas Seekamp, Ralf Baron

**Affiliations:** 1Department of Neurology, Division of Neurological Pain Research and Therapy, Universitätsklinikum Schleswig-Holstein, Campus Kiel, 24105 Kiel, Germany; 2Department of Orthopedics and Trauma Surgery, Universitätsklinikum Schleswig-Holstein, Campus Kiel, 24105 Kiel, Germany; 3Private Practice for Orthopedics and Trauma Surgery am Arndtplatz, 24116 Kiel, Germany

**Keywords:** osteoarthritis, quantitative sensory testing, total joint replacement, somatosensory phenotype, sensitization

## Abstract

To investigate sensory changes, physical function (pF), quality of life (QoL) and pain intensity of patients with osteoarthritis (OA) in the natural course of disease, and patients undergoing total joint replacement therapy (TJR) 31 (20 females, mean age 64.6 ± 10.4 years), patients with OA were investigated with questionnaires and quantitative sensory testing (QST) in the area of referred pain at the thigh at baseline and follow-up 22–49 weeks later; changes were analyzed separately for patients with (*n* = 13) and without TJR (*n* = 18). In patients without TJR pain intensity, pF, QoL did not improve, and increased pain sensitivity to cold and a stronger loss of detection were observed. In patients after TJR, however, a reduction in mechanical pain sensitivity and allodynia occurred in accordance with a reduction of pain intensity and improvement of functionality while QoL did not improve. Additionally, an increased sensitivity to heat pain and a more pronounced loss of mechanical detection could be observed in this group. TJR seems to stop peripheral pain input leading to a reduction of pain intensity and central sensitization, but surgery-induced sensory changes such as peripheral sensitization and loss of detection occur. Furthermore, TJR has favorable effects on pain intensity and functionality but not QoL.

## 1. Introduction

Osteoarthritis (OA), especially of the hip and the knee, is a frequent syndrome which leads to a considerable health problem in the afflicted patients and a substantial use of the healthcare system [1]. Amongst a huge amount of restrictions due to the disease, the pain caused by osteoarthritis is considered as being one of the main reasons causing a relevant decrease in quality of life [2]. Until now, the underlying pain mechanisms are not entirely identified, but there are hints that not only nociceptive mechanisms caused by damage to the joint tissue are causative. At that, animal experimental data as well as several clinical trials suggest that neuropathic pain components can also occur in OA [3,4,5]. Neuropathic pain is usually defined as pain caused by a lesion or disease of the somatosensory system [6]. Regarding OA, it has been demonstrated that small nociceptive nerve fiber endings normally innervating the joint capsule and the bone start sprouting into the degenerated cartilage [7]. Continuous compression of the cartilage within the arthritic joint then leads to ongoing damage of these fibers which can cause neuropathic pain. Accordingly, up to 1/3 of the patients with osteoarthritis use descriptors of their sensory perceptions that are characteristic for neuropathic pain, e.g., describing the pain as a burning or heavy sensation, or numbness [8,9]. In chronic neuropathic pain, pathophysiological mechanisms such as peripheral and central sensitization occur [10]. Peripheral sensitization is defined as an increased responsiveness and reduced threshold of nociceptive neurons in the periphery to the stimulation of their receptive fields, whereas central sensitization is defined as increased responsiveness of nociceptive neurons in the central nervous system to their normal or sub-threshold afferent input [11]. Several clinical trials showed indications for the presence of central sensitization in patients with symptomatic OA [12,13,14].

The quantitative sensory testing (QST) battery by the German research Network on Neuropathic Pain (DFNS) investigates afferent peripheral nerve fiber functions or central pathways [15] and can therefore help to detect dysfunctions of the somatosensory system that are usually present in neuropathic pain [16]. By comparing the individual measured values with normative values of healthy controls, it is able to classify individual somatosensory function for the different QST parameters as normal or abnormal [15], therewith addressing the hypothesis that different clinical signs reflect different underlying mechanisms of neuropathic pain [16]: presence of heat hyperalgesia, for example, is thought to be a result of peripheral sensitization [17], whereas mechanical allodynia and hyperalgesia are the result of a central sensitization of the somatosensory nervous system [18,19,20,21].

Joint replacement surgery is a common treatment for chronic osteoarthritis, has increased over the last 2–3 decades, and is suggested to increase further in the future [13,14]. However, conservative treatment is widely available with little treatment related side-effects, but up to now is not able to prevent disease progression [22]. There have been several studies in the past investigating physical [12] and somatosensory [3] functions as well as quality of life [2] in patients with osteoarthritis—but there has been no study investigating these aspects in the course of disease with, in particular, a focus on underlying disease mechanisms.

Aims of this study were therefore to thoroughly investigate the changes of somatosensory profile, somatosensory characteristics and pain intensity, physical function, and quality of life in patients with OA in the natural course of disease and in patients undergoing total joint replacement surgery.

## 2. Methods

### 2.1. Experimental Set-Up

Patients diagnosed with OA by orthopedic and trauma specialists were included. All patients were investigated at inclusion of the study (visit 1, baseline) as well as at follow-up (visit 2) either in the natural course of disease at least 21 weeks later or at least 19 weeks after TJR, respectively. Patients without TJR, i.e., those who were followed-up on in the natural course of disease, continued the treatment (analgesics, physiotherapy) which they already perceived at visit 1 and/or were allowed to optimize treatment in the course of disease in case of insufficiency in between visits 1 and 2 (best treatment). Treatment decisions regarding conservative treatment or joint replacement were made by the independent orthopedic or trauma specialists not associated with this study. Upon both visits, average pain intensity during the previous 3 days was documented, patients completed questionnaires (for health-related quality of life (QoL), the EuroQol-5D-3L (EQ-5D) and SF-12 Health Survey (SF-12) [23,24,25], for physical function the Hannover functional ability questionnaire for osteoarthritis (FFbH-OA) [26] and the Western Ontario and McMaster Universities Arthritis Index (WOMAC; knee osteoarthritis cohort only [27,28]), the Hospital Anxiety and Depression Scale (HADS) as a screening tool for depression or anxiety [29,30] and QST was performed. The study was in accordance with the Declaration of Helsinki and approved by the local ethics committee (AZ A153/12). All patients gave written, informed consent for participation in the study.

### 2.2. Questionnaires

#### 2.2.1. EuroQol-5D-3L (EQ-5D)

The EQ-5D is a tool to examine the subjective overall health status. It consists of two parts. The first part is a questionnaire that comprises five operational dimensions: mobility self-care, usual activities, pain/discomfort, and anxiety/depression. Each dimension has three levels: no problems, some problems, and extreme problems. In the second part, patients have to mark their overall health state on a 101-point visual analogue scale (0 = worst imaginable health state; 100 = best imaginable health state) representing a subjective rating of their health [31].

#### 2.2.2. SF-12 Health Survey (SF-12)

The SF-12 is a disease independent gauge to assess the health-related quality of life. It examines the general health as well as the subscores for physical and psychological/mental health status, respectively. Values > 60% are above average and values < 40% represent below average health status [32,33,34].

#### 2.2.3. Hannover Functional Ability Questionnaire for Osteoarthritis (FFbH-OA)

The FFbH-OA rates the physical function in patients with osteoarthritis. The score ranges from 0–100% of function. Scores from 80–100% represent normal physical function; scores from 60–80% reflect a mild impairment of physical function; scores < 60% show relevant impairment of function [26].

#### 2.2.4. Western Ontario and McMaster Universities Arthritis Index (WOMAC)

The WOMAC is a tri-dimensional, disease-specific patient reported outcome measure with three subscales assessing pain intensity, stiffness, and functional limitations resulting in one overall score. The higher the (sub-) score, the more severe the subitems pain intensity, stiffness, functional limitations due to OA, respectively [27,28].

#### 2.2.5. Hospital Anxiety and Depression Scale (HADS)

The HADS is used to screen for the presence of anxiety and depression in patients with chronic diseases [29].Two subscales represent symptoms of depression and anxiety and in each subscale a cut-off value of ≥11 is regarded as pathological, whereas a score from 8–10 is seen as borderline values indicating a unclear presence of anxiety and/or depression. The global score reflects the possible overall psychological impairment which should entail further thorough clinical diagnostics [30].

#### 2.2.6. Quantitative Sensory Testing (QST)

Since pain arising from hip OA frequently radiates to the knee, thigh, and buttocks, while knee OA is often described around the knee and on the upper tibia [35], QST at both visits was performed at the distal thigh for both hip and knee OA to minimize differences due to different areas of testing. QST was performed according to the protocol of the DFNS as described previously [15]. In short, this QST protocol consists of 13 parameters: mechanical detection threshold (MDT) and vibration detection threshold (VDT) representing function of large myelinated fibers or central pathways, cold detection threshold (CDT), cold pain threshold (CPT), warm detection threshold (WDT), heat pain threshold (HPT), thermal sensory limen (TSL), mechanical pain threshold (MPT), mechanical pain sensitivity (MPS), and pressure pain threshold (PPT) representing small fiber function or central pathways. The presence of paradoxical heat sensations (PHS) represents dysfunction of C-fibers or central pathways encoding for cold sensations [36]. Positive signs investigated include dynamic mechanical allodynia (DMA) and wind-up ratio (WUR).

Tests were always performed in the same order, starting with thermal tests. Thermal tests (CDT, WDT, CPT, HPT) were assessed using a thermode (TSA 2001-II; Medoc, City, Israel; contact area 7.84 cm^2^) placed on the skin in the testing area with a baseline temperature of 32 °C. The thermode increased/decreased its temperature by 1 °C/s until the patient detected the onset of the stimulus and clicked the response button causing the thermode to return to the baseline temperature of 32 °C. There were 3 stimuli in total, with a 10 s interval between single stimuli. An average threshold temperature was then calculated from the three assessments. To obtain the thermal sensory limen (TSL), the thermode constantly altered its temperature starting at 32 °C. Whenever a stimulus was detected, patients were asked to push the response button and rate the stimuli as being warm or cold. Upon pushing the response button, the thermode gradually reversed the direction of the temperature. Six stimuli were applied and the square mean was calculated. The number of paradoxical heat sensations during this procedure was recorded. Paradoxical heat sensations were present when a patient reported a stimulus as warm, hot or burning when in fact a cold stimulus was applied.

For MDT, a standardized set of von Frey filaments (Optihair2-Set, MARSTOCK nervtest, Schriesheim, Germany), exerting a force between 0.25 and 512 mN [37,38] was used. According to the ‘method of limits’, thresholds were assessed by stepwise increasing and decreasing stimulus intensity (up-down-rule). A geometric mean (in mN) was calculated from five ascending and descending stimulus series.

For assessment of MPT pinprick stimuli with fixed stimulus intensities (8, 16, 32, 64, 128, 256, 512 mN; The PinPrick; MRC Systems GmbH, Heidelberg, Germany) were used. For MPT, thresholds were measured by stepwise increasing and decreasing stimulus intensity. A geometric mean (in mN) was calculated from five ascending and descending stimulus series.

For the mechanical pain sensitivity (MPS) of the skin and the dynamic mechanical allodynia (DMA) pinprick, stimuli were interspersed with a paintbrush exerting a force between 200–400 mN, a Q-tip mounted on a flexible plastic ruler with a force of approximately 100 mN and a cotton wisp providing a force of approximately 3 mN stroked over 1–2 cm skin. Fifty stimuli were applied per side in a pseudo-randomized order. The patient reported the painfulness of each stimulus using a numerical rating scale (0–100, where 0 = not painful, blunt, 100 = the most intense pain imaginable). Pain elicited by stroking with normally non painful stimuli such as the Q-tip, paintbrush and cotton wisp (pain to light touch) provided evidence of dynamic mechanical allodynia [39].

WUR was defined by the perceived magnitude of pain to a series of pinprick stimuli (pinprick force: 256 mN, repeated 10 times at a 1/s rate on separate locations within a small area of about 1 cm^2^) compared to a single pinprick stimulus of the same force.

VDT was measured using a standardized tuning fork (64 Hertz Rydel–Seiffer, 8/8 Skala) placed on the patella for OA of the knee or the anterior superior iliac spine for OA of the hip. Patients indicated when they could no longer detect vibration. The vibration threshold was determined by calculating a mean of 3 repetitions.

For PPT, the tip of a pressure gauge (was applied to the skin over a muscle at the thigh and pressure continuously raised at 50 kPa/s until the patient reported pain from the pressure. The pressure values at pain threshold were recorded and an average was calculated from three repetitions.

### 2.3. Statistical Analysis

QST-results were analyzed according to published guidelines [15] and compared to a reference data base of healthy controls [40,41]. Since normative values for the thigh do not exist in the reference database, patients’ values were compared to the reference values of the dorsum of the feet of healthy controls for a calculation of z-values. Therefore, patient data were normalized to the respective gender and age group of the healthy controls and z-values calculated (z = (individual value − mean_data base_)/SD_database_). The resulting Z-scores above “0” indicate hyperfunction, i.e., patients are more sensitive to the tested parameter compared to controls (lower thresholds), whereas Z-scores below “0” indicate hypofunction and therefore a loss of or lower sensitivity of the patient compared to controls (higher thresholds). Wilcoxon-test was used for intra-individual comparison at baseline and at follow-up. *p* < 0.05 was considered statistically significant.

## 3. Results

### 3.1. Characteristics of Allpatients

Characteristics of patients are shown in Table 1. Mean pain intensity in the study sample was moderate. Thirteen (42%) patients did regularly take analgesics, 7 (22.6%) whereat 86.4% used non-opioids, 13.6% low potent opioids, and 27.3% co-analgesics. Seven (22.6%) patients used analgesics on demand (90.9% non-opioids and 9.1% co-analgesis) and 11 (35.5%) did not take any pain medication. Psychological comorbidities in the overall study sample were below 25% (Table 1). Eighteen (58%) patients scored a below average health status upon a physical sub score of SF-12 and a 6 (19.4%) below average health status upon the psychological/mental sub score of SF-12. Twenty-four (77.4%) patients reported a mild (22.5%) or relevant (51.6%) impairment of function according to the FFbH-OA. Most of the patients showed moderate impairment of health status except pain and distress upon the different sub scores of EQ-5D (Table 2).

Both groups of patients, i.e., those with and without undergoing TJR later on, did not differ regarding age, duration of symptoms, pain intensity, BMI, positive screening for anxiety and depression, WOMAC overall score and sub scores, QoL (EQ-5D, SF-12 physical and psychological/mental sub score) during visits, but patients undergoing TJR later on showed a stronger impairment upon FFbH-OA, which explains the decision for surgery in this subgroup (*p* = 0.025, Table 1).

### 3.2. Patients without TJR

A follow-up visit (visit 2) was performed 21–49 weeks (mean 34.2 ± 10.7 weeks) after visit 1. A trend towards a slight reduction of average pain intensity between visit 1 and 2 could be detected (3.7 ± 2.3 vs. 2.7 ± 2.2; p, n.s.). Maximum pain intensity (6.4 ± 2.3 vs. 7.8 ± 14.5; p, n.s.) and quality of life (Table 2) did not improve. Functionality only partly improved with an improvement in WOMAC sub scores for stiffness and functional limitations in those with knee arthritis, whereas the other questionnaires investigating functionality did not show differences of functionality (Table 2).

Upon QST, the cold pain threshold decreased and a stronger loss for mechanical detection was observed for visit 2 compared to visit 1 (Figure 1). On the individual level, however, frequencies of abnormal values did not change between visits 1 and 2.

### 3.3. Patients with TJR

Follow-up visit (visit 2) was performed 19–35 weeks (mean 26.2 ± 4.1 weeks) after visit 1. In contrast to patients without TJR, mean (3.7 ± 2.3 vs. 2.7 ± 2.2; *p* < 0.01) and maximum (8.0 ± 1.8 vs. 2.4 ± 2.8; *p* < 0.01) pain intensity decreased and functionality improved from visits 1 to 2 in patients that underwent TJR, whereas no differences in QoL were observed (Table 2). Upon QST, a reduction in increased sensitivity to mechanical pain (MPS) as well as a reduction of dynamic mechanical allodynia (DMA), but an increased sensitivity to heat pain (HPT) and a more pronounced loss for mechanical detection (MDT, VDT), were observed after TJR compared to baseline (Figure 2). As in patients without TJR, frequencies of pathological values did not change between visits 1 and 2.

## 4. Discussion

The study shows that pain intensity, functionality, and QoL in OA patients without TJR do not or only slightly improve in the course of disease. Instead, increased pain sensitivity towards cold and a stronger loss of detection can be observed. Supporting our findings, Crawford et al. described long-term conservative treatment that showed no meaningful improvement in pain relief or physical function; only short-term relief was found in some studies [22].

In OA patients after TJR, however, a reduction in parameters representing central sensitization (MPS, DMA) occurs in accordance with a reduction of pain intensity and improvement of functionality, whereas, interestingly, QoL does not improve. In contrast, an increased sensitivity to heat pain and a more pronounced loss in mechanical detection can be observed. The concomitant decrease of pain intensity and improvement of QST parameters representing central sensitization in patients who underwent TJR point towards a crucial role of pain intensity for the development and maintenance of central sensitization phenomena. Similar results have been generated by Kosek et al. [42] who demonstrated that abnormal QST values normalized after successful treatment in OA-patients. Yarnitzky et al. define a pronociceptive pain profile, a phenotype, in which patients are more likely to develop severe pain, a higher prevalence of pain syndromes, and sustaining overall more severe pain than the anti-nociceptive phenotype, which is less likely to suffer from severe, ongoing pain [43]. One might suggest that the presence of central sensitization processes might reveal a pro-nociceptive pain profile in patients with osteoarthritis. In our patient cohort, signs of central sensitization diminish after TJR and therefore the malfunction of central pain processing pathways seems to normalize possibly due to the absence of nociceptor stimulation.

For visit 2, a stronger loss of mechanical detection and an increased sensitivity to heat pain could be demonstrated, suggesting additional pathological peripheral mechanisms that occur in consequence of TJR, e.g., a lesion of cutaneous nerve fibers as a part of tissue damage due to surgery.

The main limitation of this study is the small sample size. Furthermore, treatment of patients without TJR was not documented in detail.

Interestingly, despite improvement in pain intensity and functionality after TJR, no changes in QoL parameters could be detected. As shown by Gierthmühlen et al., pain and symptom intensity, functionality, and QoL are not necessarily reported in association with one another [44]. To this point, the reasons for this observation are not clear. Maybe psycho-social factors like family and social environment as well as workplace need to be additionally taken into account.

## 5. Conclusions

Summing up, results of this study suggest that, in contrast to conservative treatment, TJR stops peripheral pain input, resulting in a reduction of pain intensity and central sensitization phenomena. Whether TJR is therefore favorable compared to conservative treatment needs to be elucidated in future studies.

## Figures and Tables

**Figure 1 ijerph-17-03085-f001:**
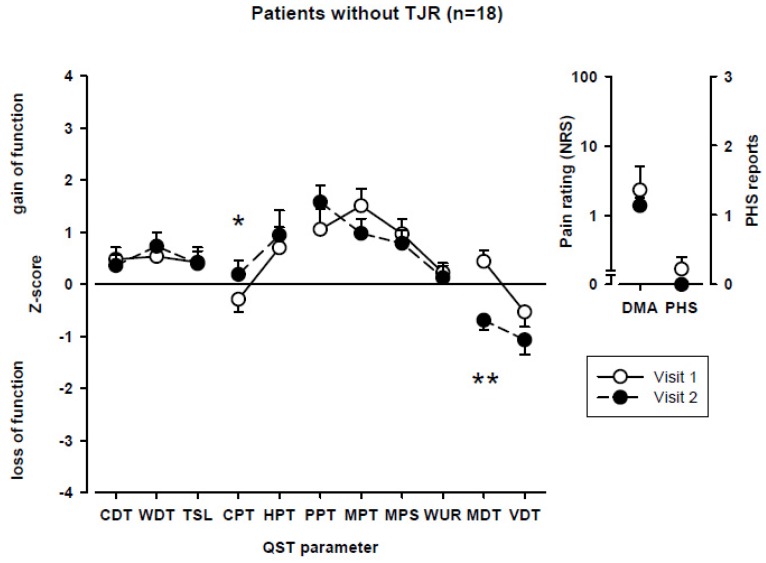
Somatosensory profiles of patients without TJR at baseline (visit 1) and follow-up visit (visit 2). CDT, cold detection threshold; WDT, warm detection threshold; TSL, thermal sensory limen; CPT, cold pain threshold; HPT, heat pain threshold; PPT, pressure pain threshold; MPT, mechanical pain threshold; MPS, mechanical pain sensitivity; WUR, wind-up ratio; MDT, mechanical detection threshold; VDT, vibration detection threshold; DMA, dynamic mechanical allodynia; PHS, paradoxical heat sensation; TJR, total joint replacement, * *p* < 0.05 patients without TJR compared for visits 1 and visit 2. ** *p* < 0.01 patients without TJR compared for visits 1 and visit 2.

**Figure 2 ijerph-17-03085-f002:**
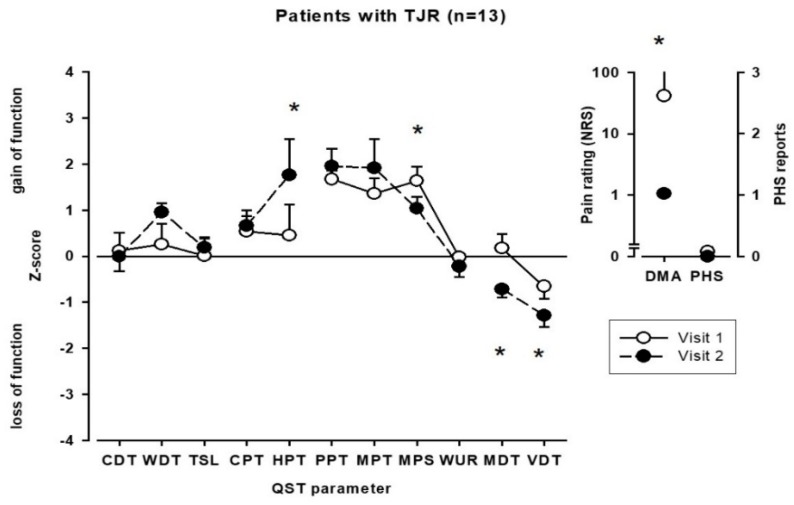
Somatosensory profiles of patients with TJR at baseline (visit 1) and follow-up visit (visit 2). CDT, cold detection threshold; WDT, warm detection threshold; TSL, thermal sensory limen; CPT, cold pain threshold; HPT, heat pain threshold; PPT, pressure pain threshold; MPT, mechanical pain threshold; MPS, mechanical pain sensitivity; WUR, wind-up ratio; MDT, mechanical detection threshold; VDT, vibration detection threshold; DMA, dynamic mechanical allodynia; PHS, paradoxical heat sensation; TJR, total joint replacement; * *p* < 0.05, patients with TJR compared for visit 1 and visit 2.

**Table 1 ijerph-17-03085-t001:** Study Population at visit 1.

	All (*n* = 31)	Osteoarthritis Patients with TJR (*n* = 13)	Osteoarthritis Patients without TJR (*n* = 18)	P (with vs. without TJR)
Age [years]	64.6 ± 10.4 (41–85)	62.9 ± 12.0 (41–85)	65.8 ± 9.3 (50–77)	n.s.
females (*n*, %)	20 (64.5%)	8 (61.4%)	12 (66.7%)	n.s.
BMI	29.6 ± 7.0 (16.8–50.2)	31.3 ± 8.5 (21.7–50.2)	28.4 ± 5.6 (16.8–43.4)	n.s.
OA of the knee (*n*, %)	20 (64.5%)	8 (61.4%)	12 (66.7%)	n.s.
Average Pain Intensity (NRS-3)	4.3 ± 2.2 (1–8.5)	5.1 ± 1.9 (1–8)	3.7 ± 2.3 (1–8.5)	n.s.
Maximum Pain Intensity	7.1 ± 2.2 (3–10)	8.0 ± 1.8 (4–10)	6.4 ± 2.3 (3–10)	n.s.
Duration of symptoms [months]	5.5 ± 6.4 (0.25–30)	4.3 ± 3.2 (1–10)	6.4 ± 8.0 (0.25–30)	n.s.
HADS positive screening for anxiety	5 (16.1%)	3 (23%)	2 (11.1%)	n.s.
HADS positive screening for depression	2 (6.5%)	1 (7.7%)	1 (5.6%)	n.s.

Values are given as mean ± SD (range). OA, osteoarthritis; NRS-3, average pain intensity within the last 3 days; TJR, Total Joint Replacement.

**Table 2 ijerph-17-03085-t002:** Results of questionnaires.

	OA Patients with TJR (V1) (*n* = 13)	OA Patients with TJR (V2) (*n* = 13)	OA Patients without TJR (V1) (*n* = 18)	OA Patients without TJR (V2) (*n* = 18)	P (with TJR V1 vs. V2)	P (without TJR V1 vs. V2)
WOMAC overall score	110.0 ± 49.7 (30–197)	54.6 ± 53.6 (6–161)	80.6 ± 50.0 (18–170)	76.8 ± 43.1 (3–140)	0.011	0.038
WOMAC pain intensity	23.9 ± 8.8 (9–38)	8.0 ± 6.0 (0–21)	15.6 ± 10.6 (0–33)	15.4 ± 9.0 (1–28)	0.008	n.s.
WOMAC stiffness	9.9 ± 6.4 (0–20)	6.4 ± 6.0 (0–20)	9.4 ± 4.9 (3–20)	6.7 ± 5.4 (0–17)	n.s.	0.008
WOMAC functional limitations	44.84 ± 22.03	23.69 ± 25.93	55.6 ± 35.9 (12–119)	53.1 ± 30.7 (2–100)	0.021	0.024
SF12: physical Mean ± SD	31.1 ± 10.4 (20.7–53.2)	39.1 ± 9.1(21.2–50.7)	31.1 ± 15.5 (0–54.6)	32.6 ± 14.7 (0–57.6)	0.023	n.s.
abnormal value (*n*, %)	10 (83.3%)	5 (38.5%)	8 (66.7%)	9 (64.3%)	0.022	n.s.
SF12: psychological Mean ± SD	48.7 ± 12.2 (27.4–67.5)	54.0 ± 9.6 (32.8–63.4)	49.1 ± 13.0 (28.9–62.1)	49.8 ± 12.1 (32.8–64.1)	n.s.	n.s.
abnormal value (*n*, %)	2 (15.4%)	3 (25.0%)	4 (28.6%)	3 (25.0%)	n.s.	n.s.
FFbH-OA Mean ± SD	50.9 ± 17.5 (25–81.25)	73.5 ± 18.8 (33.3–94.4)	67.4 ± 21.0 (25–100)	70.2 ± 19.6 (38.9–100)	0.002	n.s.
abnormal value (*n*, %)	12 (92.3%)	6 (46.2%)	12 (66.7%)	12 (66.7%)	< 0.05	n.s.
EQ-5D: mobility						
moderate impairment (*n*, %)	10 (76.9%)	8 (61.5%)	10 (55.6%)	12 (66.7%)	n.s.	n.s.
strong impairment (*n*, %)	0	0	0	0		
EQ-5D: self-sufficiency						
moderate impairment (*n*, %)	2 (15.4%)	2 (15.4%)	3 (16.7%)	0	n.s.	n.s.
strong impairment (*n*, %)	0	0	1 (5.6%)	2 (11.1%)		n.s.
**EQ-5D: general activities**						
moderate impairment (*n*,%)	10 (76.9%)	7 (53.8%)	11 (61.1%)	13 (72.2%)	n.s.	n.s.
strong impairment (*n*, %)	0	0	0	0		
**EQ-5D: pain and distress**						
moderate impairment (*n*,%)	9 (69.2%)	9 (69.2%)	10 (55.6%)	11 (61.1%)	n.s.	n.s.
strong impairment (*n*, %)	4 (30.8%)	1 (7.7%)	4 (22.2%)	3 (16.7%)	n.s.	n.s.
**EQ-5D: anxiety and depression**						
moderate impairment (*n*,%)	5 (38.5%)	4 (30.8%)	7 (38.9%)	8 (44.4%)	n.s.	n.s.
strong impairment (*n*, %)	1 (7.7%)	0	0	0	n.s.	

Mean ± SD (range). OA, osteoarthritis; TJR, total joint replacement; WOMAC, Western Ontario and McMaster Universities Arthritis Index; SF-12, Short Form 12; FFbH-OA, Hannover functional ability questionnaire for osteoarthritis; EQ-5D, EuroQol-5D-3L. n.s.: not significant.

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
