# Peer review of "Changes of Somatosensory Phenotype in the Course of Disease in Osteoarthritis Patients"

_ijerph, 2020, doi:10.3390/ijerph17093085_

Round 1
Reviewer 1 Report
This is a manuscript well written but some issues could be improved in order to achieve a better research quality
Introduction section shows and adequate background but is required a better and deep knowledge in order to get the main idea. The hipothesis could be rewrited to achieve a better and clear idea.
Methods needs a re-written in order to achieve a better understanding of the procedure. Redaction should be clear. The design is appropiate
Results sectiion is clearly showed. Congratulations.
Conclusion are clearly supported by the data.
Author Response
Reviewer #1:
This is a manuscript well written but some issues could be improved in order to achieve a better research quality.
Introduction section shows and adequate background but is required a better and deep knowledge in order to get the main idea. The hipothesis could be rewrited to achieve a better and clear idea.
Answer: We have now extended the introduction and described the background in more detail. We hope that the main idea is now clearer.
Methods needs a re-written in order to achieve a better understanding of the procedure. Redaction should be clear. The design is appropriate.
Answer: We have tried to write the methods section clearer and hope that it can now be understood better.
Results sectiion is clearly showed. Congratulations.
Answer: Thank you.
Conclusion are clearly supported by the data.
Answer: Thank you.
Reviewer 2 Report
Line 62: Timing of visit 1 is not clear: is it performed both before JRT and before beginning conservative therapy? I think it is better to specify.
Line 62-63: you wrote "follow-up at least 21 weeks later". Do you mean "after beginning conservative therapy"?
Line 74-75 Which normative values of healthy controls did you use to compare QST patient's values? References 26-27 do not contain normative values obtained from the distal thigh. Do you have your own normative datas? In this case, which was the number of healthy tested controls?
Author Response
Reviewer #2:
Line 62: Timing of visit 1 is not clear: is it performed both before JRT and before beginning conservative therapy? I think it is better to specify.
Answer: In the revised manuscript we have tried to specify that visit 1 was performed at inclusion and visit 2 at follow up either after TJR or in the course of the disease (lines 77-80) and hope that it is clearer now.
Line 62-63: you wrote "follow-up at least 21 weeks later". Do you mean "after beginning conservative therapy"?
Answer: The patients of the study were sent from orthopedics or trauma surgents for inclusion into the study (visit 1). Most of them already had a sufficient treatment which they were allowed to continue until visit 2. In case of insufficient treatment in the patients with observation of the natural course of disease, they were also allowed to optimize their treatment (in order to obtain best medical treatment). We have tried to state this point clearer in the revised manuscript.
Line 74-75 Which normative values of healthy controls did you use to compare QST patient's values? References 26-27 do not contain normative values obtained from the distal thigh. Do you have your own normative data? In this case, which was the number of healthy tested controls?
Answer: You are right that normative values for the thigh do not exist in the reference database of the DFNS. Therefore, patients’ values were compared to the reference values of the dorsum of the feet of healthy controls. We have clearly written this in the revised manuscript (lines 188-190). However, since the manuscript focusses on the follow-up examination of sensory signs using intra-individual comparisons of QST values and testing was performed in the same areas at baseline and follow-up, we think that this otherwise important point does not matter with regard to this study.
Round 2
Reviewer 2 Report
Dear authors, it is actually not possibile to compare QST results obtained from the thigh with normative data obtained from the dorsum of the feet. There are no evidences that the values obtained from this different sites are comparable. You should collect your own normative data if you want to compare your result to a healthy population. Otherwise, you can decide only to compare intra-individual values at baseline and at follow-up.
Author Response
Dear reviewer, we agree that it is actually not possible to compare QST results obtained from the thigh with normative data obtained from the dorsum of the feet and that we should collect our own normative data for future studies that compare patients with healthy controls. However, in this study – as you mentioned in your review comment- we have only compared intra- intra-individual values at baseline and at follow-up within the two study groups. We have now tried to make this clearer in the revised manuscript (green marks).
